# Model evidence from nonequilibrium simulations

**Michael Habeck**
Statistical Inverse Problems in Biophysics, Max Planck Institute for Biophysical Chemistry &
Institute for Mathematical Stochastics, University of Göttingen, 37077 Göttingen, Germany
`email` mhabeck@gwdg.de

## Abstract

The marginal likelihood, or model evidence, is a key quantity in Bayesian parameter estimation and model comparison. For many probabilistic models, computation of the marginal likelihood is challenging, because it involves a sum or integral over an enormous parameter space. Markov chain Monte Carlo (MCMC) is a powerful approach to compute marginal likelihoods. Various MCMC algorithms and evidence estimators have been proposed in the literature. Here we discuss the use of nonequilibrium techniques for estimating the marginal likelihood. Nonequilibrium estimators build on recent developments in statistical physics and are known as annealed importance sampling (AIS) and reverse AIS in probabilistic machine learning. We introduce estimators for the model evidence that combine forward and backward simulations and show for various challenging models that the evidence estimators outperform forward and reverse AIS.

## 1 Introduction

The marginal likelihood or model evidence is a central quantity in Bayesian inference [1, 2], but notoriously difficult to compute. If likelihood $L(x) \equiv p(y|x, M)$ models data $y$ and prior $\pi(x) \equiv p(x|M)$ expresses our knowledge about the parameters $x$ of the model $M$, the posterior $p(x|y, M)$ and the model evidence $Z$ are given by:

$$p(x|y, M) = \frac{p(y|x, M)\, p(x|M)}{p(y|M)} = \frac{L(x)\,\pi(x)}{Z}, \quad Z \equiv p(y|M) = \int L(x)\,\pi(x)\, dx\,. \quad (1)$$

Parameter estimation proceeds by drawing samples from $p(x|y, M)$, and different ways to model the data are ranked by their evidence. For example, two models $M_1$ and $M_2$ can be compared via their Bayes factor, which is proportional to the ratio of their marginal likelihoods $p(y|M_1)/p(y|M_2)$ [3].

Often the posterior (and perhaps also the prior) is intractable in the sense that it is not possible to compute the normalizing constant and therefore also the evidence analytically. This makes it difficult to compare different models via their posterior probability and model evidence. Markov chain Monte Carlo (MCMC) algorithms [4] only require unnormalized probability distributions and are among the most powerful and accurate methods to estimate the marginal likelihood, but they are computationally expensive. Therefore, it is important to develop efficient MCMC algorithms that can sample from the posterior and allow us to compute the marginal likelihood.

There is a close analogy between the marginal likelihood and the log-partition function or free energy from statistical physics [5]. Therefore, many concepts and algorithms originating in statistical physics have been applied to problems arising in probabilistic inference. Here we show that nonequilibrium fluctuation theorems (FTs) [6, 7, 8] can be used to estimate the marginal likelihood from forward and reverse simulations.

## 2 Marginal likelihood estimation by annealed importance sampling

A common MCMC strategy to sample from the posterior and estimate the evidence is to simulate a sequence of distributions $p_k$ that bridge between the prior and the posterior [9]. Samples can either be generated in sequential order as in annealed importance sampling (AIS) [10] or in parallel as in replica-exchange Monte Carlo or parallel tempering [11, 12]. Several methods have been proposed to estimate the marginal likelihood from MCMC samples including thermodynamic integration (TI) [9], annealed importance sampling (AIS) [10], nested sampling (NS) [13] or the density of states (DOS) [14]. Most of these approaches (TI, DOS and NS) assume that we can draw exact samples from the intermediate distributions $p_k$, typically after a sufficiently large number of equilibration steps has been simulated. AIS, on the other hand, does not assume that the samples are equilibrated after each annealing step, which makes AIS very attractive for analyzing complex models for which equilibration is hard to achieve.

AIS employs a sequence of $K + 1$ probability distributions $p_k$ and Markov transition operators $T_k$ whose stationary distributions are $p_k$, i.e. $\int T_k(x|x')\, p_k(x')\, dx' = p_k(x)$. In a Bayesian setting, $p_0$ is the prior and $p_K$ the posterior. Typically, $p_k$ is intractable meaning that we only know an unnormalized version $f_k$, but not the normalizer $Z_k$, i.e. $p_k(x) = f_k(x)/Z_k$ where $Z_k = \int f_k(x)\, dx$ and the evidence is $Z = Z_K/Z_0$. Often, it is convenient to write $f_k$ as an energy based model $f_k(x) = \exp\{-E_k(x)\}$. In Bayesian inference, a popular choice is

$$f_k(x) = [L(x)]^{\beta_k}\, \pi(x)$$

with prior $\pi(x)$ and likelihood $L(x)$; $\beta_k$ is an inverse temperature schedule that starts at $\beta_0 = 0$ (prior) and ends at $\beta_K = 1$ (posterior).

AIS samples paths $\boldsymbol{x} = [x_0, x_1, \ldots, x_{K-1}]$ according to the probability

$$\mathcal{P}_f[\boldsymbol{x}] = T_{K-1}(x_{K-1}|x_{K-2})\cdots T_1(x_1|x_0)\, p_0(x_0) \tag{2}$$

where, following Crooks [8], calligraphic symbols and square brackets denote quantities that depend on the entire path. The subscript indicates that the path is generated by a *forward* simulation, which starts from $p_0$ and propagates the initial state through a sequence of new states by the successive action of the Markov kernels $T_1, T_2, \ldots, T_{K-1}$.

The importance weight of a path is

$$w[\boldsymbol{x}] = \prod_{k=0}^{K-1} \frac{f_{k+1}(x_k)}{f_k(x_k)} = \exp\left\{-\sum_{k=0}^{K-1} [\, E_{k+1}(x_k) - E_k(x_k)\, ]\right\}. \tag{3}$$

The average weight over many paths is a consistent and unbiased estimator of the model evidence $Z = Z_K/Z_0$, which follows from [15, 10] (see supplementary material for details):

$$\langle w \rangle_f = \int w[\boldsymbol{x}]\, \mathcal{P}_f[\boldsymbol{x}]\, \mathcal{D}[\boldsymbol{x}] = Z \tag{4}$$

where the average $\langle \cdot \rangle_f$ is an integral over all possible paths generated by the forward sequence of transition kernels ($\mathcal{D}[\boldsymbol{x}] = dx_0 \cdots dx_{K-1}$). The average weight of $M$ forward paths $\boldsymbol{x}^{(i)}$ is an estimate of the model evidence: $Z \approx \frac{1}{M} \sum_i w[\boldsymbol{x}^{(i)}]$. This estimator is at the core of AIS and its variants [10, 16]. To avoid overflow problems, it will be numerically more stable to compute log weights. Logarithmic weights also naturally occur from a physical perspective where $-\log w[\boldsymbol{x}]$ is identified as the *work* required to generate path $\boldsymbol{x}$.

## 3 Nonequilibrium fluctuation theorems

Nonequilibrium fluctuations theorems (FTs) quantify the degree of irreversibility of a stochastic process by relating the probability of generating a path by a forward simulation to the probability of generating the exact same path by a time-reversed simulation. If the Markov kernels $T_k$ satisfy detailed balance, time reversal is achieved by applying the same sequence of kernels in reverse order. For Markov kernels not satisfying detailed balance, the definition is slightly more general [7, 10]. Here we assume that all kernels $T_k$ satisfy detailed balance, which is valid for Markov chains based on the Metropolis algorithm and its variants [4].

Under these assumptions, the probability of generating the path $\boldsymbol{x}$ by a reverse simulation starting in $x_{K-1}$ is

$$\mathcal{P}_r[\boldsymbol{x}] = T_1(x_0|x_1)\cdots T_{K-1}(x_{K-2}|x_{K-1})\,p_K(x_{K-1})\,. \qquad (5)$$

Averages over the reverse paths are indicated by $\langle\cdot\rangle_r$. The detailed fluctuation theorem [6, 8] relates the probabilities of generating $\boldsymbol{x}$ in a forward and reverse simulation (see supplementary material):

$$\frac{\mathcal{P}_f[\boldsymbol{x}]}{\mathcal{P}_r[\boldsymbol{x}]} = \frac{Z_K}{Z_0}\prod_{k=0}^{K-1}\frac{f_k(x_k)}{f_{k+1}(x_k)} = \frac{Z}{w[\boldsymbol{x}]} = \exp\{\mathcal{W}[\boldsymbol{x}] - \Delta F\} \qquad (6)$$

where the physical analogs of the path weight and the marginal likelihood were introduced, namely the work $\mathcal{W}[\boldsymbol{x}] = -\log w[\boldsymbol{x}] = \sum_k [\, E_{k+1}(x_k) - E_k(x_k)\,]$ and the free energy difference $\Delta F = -\log Z = -\log(Z_K/Z_0)$. Various demonstrations of relation (6) exist in the physics and machine learning literature [6, 7, 8, 10, 17].

Lower and upper bounds sandwiching the log evidence [17, 18, 16] follow directly from equation (6) and the non-negativity of the relative entropy:

$$D_{\mathrm{KL}}(\mathcal{P}_f\|\mathcal{P}_r) = \int \mathcal{P}_f[\boldsymbol{x}]\log(\mathcal{P}_f[\boldsymbol{x}]/\mathcal{P}_r[\boldsymbol{x}])\,\mathcal{D}[\boldsymbol{x}] = \langle\mathcal{W}\rangle_f - \Delta F \geq 0\,.$$

From $D_{\mathrm{KL}}(\mathcal{P}_r\|\mathcal{P}_f) \geq 0$ we obtain an upper bound on $\log Z$, such that overall we have:

$$\langle\log w\rangle_f = -\langle\mathcal{W}\rangle_f \leq \log Z \leq -\langle\mathcal{W}\rangle_r = \langle\log w\rangle_r\,. \qquad (7)$$

Grosse *et al.* use these bounds to assess the convergence of bidirectional Monte Carlo [18].

Thanks to the detailed fluctuation theorem (Eq. 6), we can also relate the marginal distributions of the work resulting from many forward and reverse simulations:

$$p_f(W) = \int \delta(W - \mathcal{W}[\boldsymbol{x}])\,\mathcal{P}_f[\boldsymbol{x}]\,\mathcal{D}[\boldsymbol{x}] = p_r(W)\,e^{W-\Delta F} \qquad (8)$$

which is Crooks' fluctuation theorem (CFT) [7]. CFT tells us that the work distributions $p_f$ and $p_r$ cross exactly at $W = \Delta F$. Therefore, by plotting histograms of the work obtained in forward and backward simulations we can read off an estimate for the negative log evidence.

The Jarzynski equality (JE) [15] follows directly from CFT due to the normalization of $p_r$:

$$\int p_f(W)\,e^{-W}\,dW = \langle e^{-W}\rangle_f = e^{-\Delta F}\,. \qquad (9)$$

JE restates the AIS estimator $\langle w\rangle_f = Z$ (Eq. 4) in terms of the physical quantities. Remarkably, JE holds for any stochastic dynamics bridging between the initial and target distribution. This suggests to use fast annealing protocols to drag samples from the prior into the posterior. However, the JE involves an *exponential* average in which paths requiring the least work contribute most strongly. These paths correspond to work realizations that reside in the left tail of $p_f$. With faster annealing, the chance of generating a minimal work path decreases exponentially and becomes a rare event.

A key feature of CFT and JE is that they do not require exact samples from the stationary distributions $p_k$, which is needed in applications of TI or DOS. For complex probabilistic models, the states generated by the kernels $T_k$ will typically "lag behind" due to slow mixing, especially near phase transitions. The $k$-th state of the forward path will follow the intermediate distribution

$$q_k(x_k) = \int \prod_{l=1}^{k} T_l(x_l|x_{l-1})\,p_0(x_0)\,dx_0\cdots dx_{k-1}, \quad q_0(x) = p_0(x)\,. \qquad (10)$$

Unless the transition kernels $T_k$ mix very rapidly, $q_k \neq p_k$ for $k > 0$.

Consider the common case in Bayesian inference where $E_k(x) = \beta_k E(x)$ and $E(x) = -\log L(x)$. Then, according to inequalities (7), we have the following lower bound on the marginal likelihood

$$\langle\log w\rangle_f = -\langle\mathcal{W}\rangle_f = -\sum_{k=0}^{K-1}(\beta_{k+1} - \beta_k)\,\langle E\rangle_{q_k} \qquad (11)$$

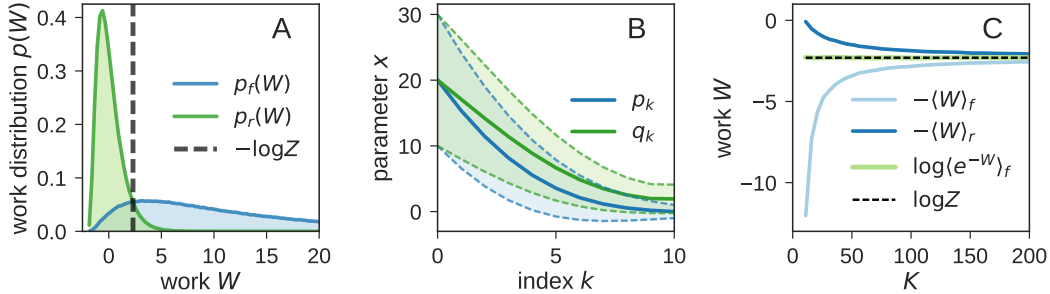

Figure 1: Nonequilibrium analysis of a Gaussian toy model. **(A)** Work distributions $p_f$ and $p_r$ shown in blue and green. The correct free energy difference (minus log evidence) is indicated by a dashed line. **(B)** Comparison of stationary distribution $p_k$ and marginal distributions $q_k$ generated by the transition kernels. Shown is a $1\sigma$ band about the mean positions for $p_k$ (blue) and $q_k$ (green). **(C)** Lower and upper bounds of the log evidence (Eq. 7) and logarithm of the exponential average over the forward work distribution for increasingly slow annealing schedules.

and an analogous expression for the upper bound/reverse direction, in which the average energies along the forward path $\langle E \rangle_{q_k}$ need to be replaced by the corresponding average energies along the backward path. The difference between the forward and reverse averages is called "hysteresis" in physics. The larger the hysteresis, the more strongly will the marginal likelihood bounds disagree and the more uncertain will our guess of the model evidence be. The opposite limiting case is slow annealing and full equilibration where the bound (Eq. 11) approaches thermodynamic integration (see supplementary material). So we expect that there is a tradeoff between switching fast in order to save computation time, versus a desire to control the amount of hysteresis, which otherwise makes it difficult to extract accurate evidence estimates from the simulations.

## 4    Illustration for a tractable model

Let us illustrate the nonequilibrium results for a tractable model where the initial, the target and all intermediate distribution are Gaussians $p_k(x) = \mathcal{N}(x; \mu_k, \sigma_k^2)$ with means $\mu_k$ and standard deviations $\sigma_k > 0$. The transition kernels are also Gaussian:

$$T_k(x|x') = \mathcal{N}\big(x; (1 - \tau_k)\mu_k + \tau_k x', (1 - \tau_k^2)\sigma_k^2\big)$$

with $\tau_k \in [0, 1]$ controlling the speed of convergence: For $\tau_k = 0$ convergence is immediate, whereas for $\tau_k \to 1$, the chain generated by $T_k$ converges infinitely slowly. Note that the kernels $T_k$ satisfy detailed balance, therefore backward simulations apply the same kernels in reverse order. The energies and exact log partition functions are $E_k(x) = \frac{1}{2\sigma_k^2}(x - \mu_k)^2$ and $\log Z_k = \log(2\pi\sigma_k^2)/2$.

We bridge between an initial distribution with mean $\mu_0 = 20$ and standard deviation $\sigma_0 = 10$ and a target with $\mu_K = 0, \sigma_K = 1$ using $K = 10$ intermediate distributions and compute work distributions resulting from forward/backward simulations. Both distributions indeed cross at $W = -\log Z = \log(\sigma_0/\sigma_K) = \log 10$, as predicted by CFT (see Fig. 1A). Figure 1B illustrates the difference between the marginal distribution of the samples after $k$ annealing steps $q_k$ (Eq. 10) and the stationary distribution $p_k$. The marginal distributions $q_k$ are also Gaussian, but their means and variances diverge from the means and variances of the stationary distributions $p_k$. This divergence results in hysteresis, if the annealing process is forced to progress very rapidly without equilibrating the samples (quenching). Figure 1C confirms the validity of the JE (Eq. 9) and of the lower and upper bounds (Eq. 7). For short annealing protocols, the bounds are very conservative, whereas the Jarzynski equality gives the correct evidence even for fast protocols (small $K$). In realistic applications, however, we cannot compute the work distribution $p_f$ over the entire range of work values. In fast annealing simulations, it will become increasingly difficult to explore the left tail of the work distribution, such that in practice the accuracy of the JE estimator deteriorates for too small $K$.

---
**Algorithm 1** Bennett's acceptance ratio (BAR)
---
**Require:** Work $W_f^{(i)}, W_r^{(i)}$ from $M$ forward and reverse nonequilibrium simulations, tolerance $\delta$ (e.g. $\delta = 10^{-4}$)
$Z \leftarrow \frac{1}{M} \sum_i \exp\{-W_f^{(i)}\}$ (Jarzynski estimator)
**repeat**

$LHS \leftarrow \sum_i \frac{1}{1 + Z \, \exp\{W_f^{(i)}\}}, \, RHS \leftarrow \sum_i \frac{1}{1 + Z^{-1} \, \exp\{-W_r^{(i)}\}}$
$Z \leftarrow Z \times \frac{LHS}{RHS}$

**until** $|\log(LHS/RHS)| < \delta$
**return** $Z$
---

## 5 Using the fluctuation theorem to estimate the evidence

To use the fluctuation theorem for evidence estimation, we run two sets of simulations. As in AIS, forward simulations start from a prior sample which is successively propagated by the transition kernels $T_k$. For each forward path $\boldsymbol{x}^{(i)}$ the total work $W_f^{(i)}$ is recorded. We also run reverse simulations starting from a posterior sample. For complex inference problems it is generally impossible to generate an exact sample from the posterior. However, in some cases the mode of the posterior is known or powerful methods for locating the posterior maximum exist. We can then generate a posterior sample by applying the transition operator $T_K$ many times starting from the posterior mode. The reverse simulations could also be started from the final states generated by the forward simulations drawn according to their importance weights $w_f^{(i)} \propto \exp\{-W_f^{(i)}\}$. Another possibility to generate a posterior sample is to start from the data, if we want to evaluate an intractable generative model such as a restricted Boltzmann machine. The posterior sample is then propagated by the reverse chain of transition operators. Again, we accumulate the total work generated by the reverse simulation $W_r^{(i)}$. The reverse simulation corresponds to reverse AIS proposed by Burda *et al.* [16].

### 5.1 Jarzynski and cumulant estimators

There are various options to estimate the evidence from forward and backward simulations. We can apply the Jarzynski equality to $W_f^{(i)}$ and $W_r^{(i)}$, which corresponds to the estimators used in AIS [10] and reverse AIS [16]. According to Eq. (7) we can also compute an interval that likely contains the log evidence, but typically this interval will be quite large. Hummer [19] has developed estimators based on the cumulant expansion of $p_f$ and $p_r$:

$$\log Z \approx -\langle \mathcal{W} \rangle_f + \mathrm{var}_f(\mathcal{W})/2, \quad \log Z \approx -\langle \mathcal{W} \rangle_r - \mathrm{var}_r(\mathcal{W})/2 \qquad (12)$$

where $\mathrm{var}_f(\mathcal{W})$ and $\mathrm{var}_r(\mathcal{W})$ indicate the sample variances of the work generated during the forward and reverse simulations. The cumulant estimators increase/decrease the lower/upper bound of the log evidence (Eq. 7) by the sample variance of the work. The forward and reverse cumulant estimators can also be combined into a single estimate [19]:

$$\log Z \approx -\frac{1}{2}\big(\langle \mathcal{W} \rangle_f + \langle \mathcal{W} \rangle_r\big) + \frac{1}{12}\big(\mathrm{var}_f(\mathcal{W}) - \mathrm{var}_r(\mathcal{W})\big). \qquad (13)$$

### 5.2 Bennett's acceptance ratio

Another powerful method is Bennett's acceptance ratio (BAR) [20, 21], which is based on the observation that according to CFT (Eq. 8)

$$\int h(W; \Delta F) \, p_f(W) \, e^{-W} \, dW = \int h(W; \Delta F) \, p_r(W) \, e^{-\Delta F} \, dW$$

for any function $h$. Therefore, any choice of $h$ gives an implicit estimator for $\Delta F$. Bennett showed [20, 9] that the minimum mean squared error is achieved for $h \propto (p_f + p_r)^{-1}$, leading to the implicit

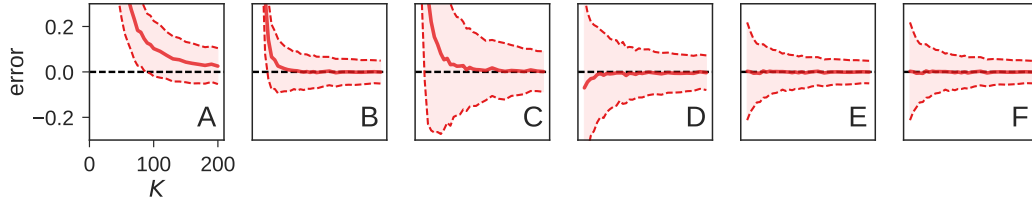

Figure 2: Performance of evidence estimators on the Gaussian toy model. $M = 100$ forward and reverse simulations were run for schedules of increasing length $K$. This experiment was repeated 1000 times to probe the stability of the estimators. Shown is the difference between the log evidence estimate and its true value $-\log 10$. The average over all repetitions is shown as red line; the light band indicates one standard deviation over all repetitions. **(A)** Cumulant estimator (Eq. 12) based on the forward simulation. **(B)** The combined cumulant estimator (Eq. 13). **(C)** Forward AIS estimator. **(D)** Reverse AIS. **(E)** BAR. **(F)** Histogram estimator.

equation

$$\sum_i \frac{1}{1 + Z \, \exp\{W_f^{(i)}\}} = \sum_i \frac{1}{1 + Z^{-1} \, \exp\{-W_r^{(i)}\}} \,. \tag{14}$$

By numerically solving equation (14) for $Z$, we obtain an estimator of the evidence based on Bennett's acceptance ratio (BAR). A straightforward way to solve the BAR equation is to iterate over the multiplicative update $Z^{(t+1)} \leftarrow Z^{(t)} \mathrm{LHS}(Z^{(t)})/\mathrm{RHS}(Z^{(t)})$ where LHS and RHS are the left and right hand side of equation (14) and the superscript $(t)$ indicates an iteration index. Algorithm (1) provides pseudocode to compute the BAR estimator (further details are given in the supplementary material).

## 5.3 Histogram estimator

Here we introduce yet another way of combining forward/backward simulations and estimating the model evidence. According to CFT, we have:

$$W_f^{(i)} \sim p_f(W), \quad W_r^{(i)} \sim p_r(W) = p_f(W) \, e^{-W}/Z \,.$$

The idea is to combine all samples $W_f^{(i)}$ and $W_r^{(i)}$ to estimate $p_f$, from which we can then obtain the evidence by using the JE (Eq. 9). Thanks to the CFT, the samples from the reverse simulation contribute most strongly to the integral in the JE. Therefore, if we can use the reverse paths to estimate the forward work distribution, $p_f$ will be quite accurate in the region that is most relevant for evaluating JE.

To estimate $p_f$ from $W_f^{(i)}$ and $W_r^{(i)}$ is mathematically equivalent to estimating the density of states (DOS) (i.e. the marginal distribution of the log likelihood) from *equilibrium* simulations run at two inverse temperatures $\beta = 0$ and $\beta = 1$. We can therefore directly apply histogram techniques [14, 22] used to analyze equilibrium simulations to estimate $p_f$ from nonequilibrium simulations (details are given in the supplementary material). Histogram techniques result in a non-parametric estimate of the work distribution:

$$p_f(W) \approx \sum_j p_j \, \delta(W - W_j) \tag{15}$$

where all sampled work values, $W_f^{(i)}$ and $W_r^{(i)}$, were combined into a single set $W_j$ and $p_j$ are normalized weights associated with each $W_j$. Using the JE, we obtain

$$Z \approx \sum_j p_j \, e^{-W_j} \tag{16}$$

which is best evaluated in log space. The histogram iterations [14] used to determine $p_j$ and $Z$ are very similar to the multiplicative that solve the BAR equation (Eq. 14). After running the histogram iterations, we obtain a non-parametric maximum likelihood estimate of $p_f$ (Eq. 15). It is also possible

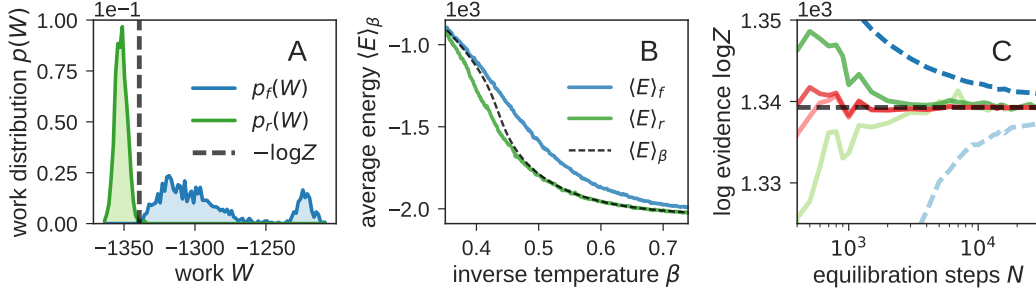

Figure 3: Evidence estimation for the $32 \times 32$ Ising model. **(A)** Work distributions obtained for $K = 1000$ annealing and $N = 1000$ equilibration steps. **(B)** Average energy $\langle E \rangle_f$ and $\langle E \rangle_r$ at different annealing steps $k$ in comparison to the average energy of the stationary distribution $\langle E \rangle_\beta$. Shown is a zoom into the inverse temperature range from 0.4 to 0.7; the average energies agree quite well outside this interval. **(C)** Evidence estimates for increasing number of equilibration steps $N$. Light/dark blue: lower/upper bounds $\langle \log w \rangle_f$ / $\langle \log w \rangle_r$; light/dark green: forward/reverse AIS estimators $\log \langle w \rangle_f$ / $\log \langle w \rangle_r$; light red: BAR; dark red: histogram estimator. For $N > 1000$, BAR and the histogram estimator produce virtually identical evidence estimates.

to carry out a Bayesian analysis, and derive a Gibbs sampler for $p_f$, which does not only provide a point estimate for $\log Z$, but also quantifies its uncertainty (see supplementary material for details).

We studied the performance of the evidence estimators on forward/backward simulations of the Gaussian toy model. The cumulant estimators (Figs. 2A, B) are systematically biased in case of rapid annealing (small $K$). The combined cumulant estimator (Fig. 2B) is a significant improvement over the forward estimator, which does not take the reverse simulation data into account. The forward and reverse AIS estimators are shown in Figs. 2C and 2D. For this system, the evidence estimates derived from the reverse simulation are systematically more accurate than the AIS estimate based on the forward simulation, which is clear given that the work distribution from reverse simulations $p_r$ is much more concentrated than the forward work distribution $p_f$ (see Fig. 1A). The most accurate, least biased and most stable estimators are BAR (Fig. 2E) and the histogram estimator (Fig. 2F), which both combine forward and backward simulations into a single evidence estimate.

# 6   Experiments

We studied the performance of the nonequilibrium marginal likelihood estimators on various challenging probabilistic models including Markov random fields and Gaussian mixture models. A python package implementing the work simulations and evidence estimators can be downloaded from `https://github.com/michaelhabeck/paths`.

## 6.1   Ising model

Our first test system is a $32 \times 32$ Ising model for which the log evidence can be computed exactly: $\log Z = 1339.27$ [23]. A single configuration consists of 1024 spins $x_i = \pm 1$. The energies of the intermediate distributions are $E_k(x) = \beta_k \sum_{i \sim j} x_i x_j$ where $i \sim j$ indicates nearest neighbors on a 2D square lattice. We generate $M = 1000$ forward and reverse paths using a linear inverse temperature schedule that interpolates between $\beta_0 = 0$ and $\beta_K = 1$ where $K = 1000$. Forward simulations start from random spin configurations. For the reverse simulations, we start in one of the two ground states with all spins either $-1$ or $+1$. $T_k$ are Metropolis kernels based on $p_k$: a new spin configuration is proposed by flipping a randomly selected spin and accepted or rejected according to Metropolis' rule. The single spin-flip transitions are repeated $N$ times at constant $\beta_k$, i.e. $N$ is the number of equilibration steps after a perturbation was induced by lowering the temperature. The larger $N$, the more time we allow the simulation to equilibrate, and the closer will $q_k$ be to $p_k$.

Figure 3A shows the work distributions obtained with $N = 1000$ equilibration steps per temperature perturbation. Even though the forward and reverse work distributions overlap only weakly, the

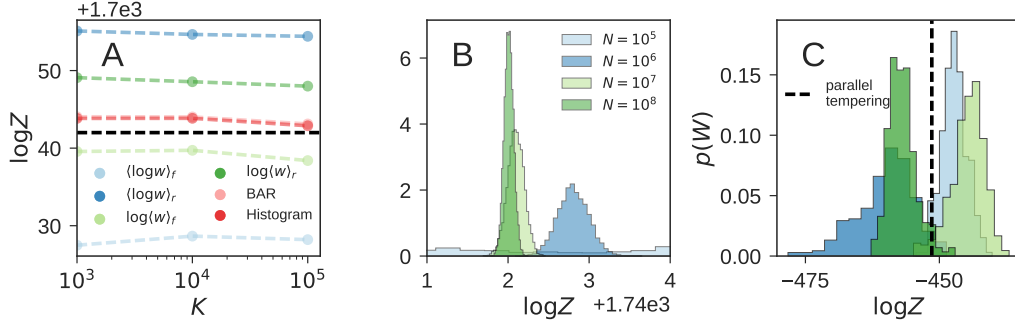

Figure 4: Evidence estimation for the Potts model and RBM. **(A)** Estimated log evidence of the Potts model for a fixed computational budget $K \times N = 10^9$ where $M = 100$ and ten repetitions were computed. The reference value $\log Z = 1742$ (obtained with parallel tempering) is shown as dashed black line. **(B)** $\log Z$ distributions obtained with the Gibbs sampling version of the histogram estimator for $K = 1000$ and varying number of equilibration steps. **(C)** Work distributions obtained for a marginal and full RBM (light/dark blue: forward/reverse simulation of the marginal model; light/dark green: forward/reverse simulation of the full model).

evidence estimates obtained with BAR and the histogram estimator are quite accurate with 1338.05 (BAR) and 1338.28 (histogram estimator), which differs only by approx. 1 nat from the true evidence and corresponds to a relative error of $\sim 9 \times 10^{-4}$ (BAR) and $7 \times 10^{-4}$ (histogram estimator). Forward and reverse AIS provide less accurate estimates of the log evidence: 1333.66 (AIS) and 1342.05 (RAISE). The lower and upper bounds are very broad $\langle \log w \rangle_f = 1290.5$ and $\langle \log w \rangle_r = 1352.0$, which results from hysteresis effects. Figure 3B zooms into the average energies obtained during the forward and reverse simulations and compares them with the average energy of a fully equilibrated simulation. The average energies differ most strongly at inverse temperatures close to the critical value $\beta_{\text{crit}} \approx 0.44$ at which the Ising model undergoes a second-order phase transition. We also tested the performance of the estimators as a function of the number of equilibration steps $N$. As already observed for the Gaussian toy model, BAR and the histogram estimator outperform the Jarzynski estimators (AIS and RAISE) also in case of the Ising model (see Fig. 3C).

### 6.2 Ten-state Potts model

Next we performed simulations of the ten-state Potts model defined over a $32 \times 32$ lattice where the spins of the Ising model are replaced by integer colors $x_i \in \{1, \ldots, 10\}$ and an interaction energy $2\delta(x_i, x_j)$. This model is significantly more challenging than the Ising model, because it undergoes a first-order phase transition and has an astronomically larger number of states ($10^{1024}$ colorings rather than $2^{1024}$ spin configurations). We performed forward/backward simulations using a linear inverse temperature schedule with $\beta_0 = 0$, $\beta_K = 1$ and a fixed computational budget $K \times N = 10^9$. Figure 4A shows that there seems to be no advantage of increasing the number of intermediate distributions at the cost of reducing the number of equilibration steps. Again, BAR and the histogram estimator perform very similarly. The Gibbs sampling version of the histogram estimator also provides the posterior of $\log Z$ (see Fig. 4B). For too few equilibration steps $N$, this distribution is rather broad or even slightly biased, but for large $N$ the $\log Z$ posterior concentrates around the correct log evidence.

### 6.3 Restricted Boltzmann machine

The restricted Boltzmann machine (RBM) is a common building block of deep learning hierarchies. RBM is an intractable MRF with bipartite interactions: $E(v, h) = -(a^T v + b^T h + v^T W h)$ where $a, b$ are the visible and hidden biases and $W$ are the couplings between the visible and hidden units $v_i$ and $h_j$. Here we compare annealing of the full model $E_k(v, h) = \beta_k E(v, h)$ against annealing of the marginal model $E_k(h) = -\beta_k \log \sum_v \exp\{-E(v, h)\}$. The full model can be simulated using a Gibbs sampler, which is straightforward since the conditional distributions are Bernoulli. To sample from the marginal model, we use a Metropolis kernel similar to the one used for the Ising model. To start the reverse simulations, we randomly pick an image from the training set and generate an initial

hidden state by sampling from the conditional distribution $p(h|v)$. We then run 100 steps of Gibbs sampling with $T_K$ to obtain a posterior sample.

We ran tests on an RBM with 784 visible and 500 hidden units trained on the MNIST handwritten digits dataset [24] with contrastive divergence using 25 steps [25]. Since the true log evidence is not known, we use a reference value obtained with parallel tempering (PT): $\log Z \approx 451.42$. Figure 4C compares evidence estimates based on annealing simulations of the full against the marginal model. Both annealing approaches provide very similar evidence estimates 451.43 (full model) and 451.48 (marginal model) that are close to the PT result. However, simulation of the marginal model is three times faster compared to the full model. Therefore, it seems beneficial to evaluate and train RBMs based on sampling and annealing of the marginal model $p(h)$ rather than the full model $p(v, h)$.

### 6.4 Gaussian mixture model

Finally, we consider a sort of "data annealing" strategy in which independent data points are added one-by-one as in sequential Monte Carlo [10] $E_k(x) = -\sum_{l<k} \log p(y_l|x, M)$. We applied thermal and data annealing to a three-component Gaussian mixture model with means -5, 0, 5, standard deviations 1, 3, 0.5 and equal weights. We generated $K = 100$ data points, and applied both types of annealing to estimate the mixture parameters and marginal likelihood. Parallel tempering produced a reference log evidence of -259.49. A Gibbs sampler utilizing cluster responsibilities as auxiliary variables served as transition kernel. Forward simulations started from prior samples where conjugate priors were used for the component means, widths and weights. The reverse simulations started from a posterior sample obtained by running K-means followed by 100 Gibbs sampling iterations.

Thermal annealing with as many temperatures as data points and 10 Gibbs sampling steps per temperature estimated a log evidence of -259.72 $\pm$ 0.60 ($M = 100$, 10 repetitions). For 100 Gibbs steps, we obtain -259.47 $\pm$ 0.36. Data annealing with 10 Gibbs steps per addition of a data point yields –257.52 $\pm$ 0.97, which seems to be slightly biased. Increasing the number of Gibbs steps to 100 improves the accuracy of the log evidence estimate: -258.32 $\pm$ 1.21. This shows that there might be some potential in a data annealing strategy, especially for larger datasets.

## 7 Summary

This paper applies nonequilibrium techniques to estimate the marginal likelihood of an intractable probabilistic model. We outline the most important results from nonequilibrium statistical physics that are relevant to marginal likelihood estimation and relate them to machine learning algorithms such as AIS [10], RAISE [16] and bidirectional Monte Carlo [17, 18]. We introduce two estimators, BAR and the histogram estimator, that are currently not used in the context of probabilistic inference. We study the performance of the estimators on a toy system and various challenging probabilistic models including Markov random fields and Gaussian mixture models. The two evidence estimators perform very similarly and are superior to forward/reverse AIS and the cumulant estimators. Compared to BAR, the histogram estimator has the additional advantage that it also quantifies the uncertainty of the evidence estimate.

### Acknowledgments

This work has been funded by the Deutsche Forschungsgemeinschaft (DFG) SFB 860, subproject B09.

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
