[Supplementary Material · nips_supplement.pdf]

# Supplementary Material:
# Model evidence from nonequilibrium simulations

**Michael Habeck**
Statistical Inverse Problems in Biophysics, Max Planck Institute for Biophysical Chemistry &
Institute for Mathematical Stochastics, University of Göttingen, 37077 Göttingen, Germany
`email` mhabeck@gwdg.de

## 1 Path simulations

The probability of simulating a path $\boldsymbol{x} = [x_0, \ldots, x_{K-1}]$ in a forward simulation is

$$\mathcal{P}_f[\boldsymbol{x}] = \prod_{k=1}^{K-1} T_k(x_k|x_{k-1}) \, p_0(x_0) \,.$$

The probability of generating the same path in a reverse simulation is:

$$\mathcal{P}_r[\boldsymbol{x}] = T_1(x_0|x_1) \cdots T_{K-1}(x_{K-2}|x_{K-1}) \, p_K(x_{K-1}) = \prod_{k=1}^{K-1} T_k(x_{k-1}|x_k) \, p_K(x_{K-1}) \,.$$

The weight of a path is

$$w[\boldsymbol{x}] = \prod_{k=0}^{K-1} \frac{f_{k+1}(x_k)}{f_k(x_k)} \,.$$

where $p_k(x) = f_k(x)/Z_k$ with $Z_k = \int f_k(x) \, dx$ is the stationary distribution of $T_k$.

### 1.1 Jarzynski equality

The Jarzynski equaliy (JE) states:

$$Z = \langle w \rangle_f \tag{1}$$

where $\langle \cdot \rangle_f$ indicates an average over paths generated in forward simulations according to $\mathcal{P}_f$.

JE follows from [1, 2]

$$
\begin{aligned}
\langle w \rangle_f &= \int w[\boldsymbol{x}] \, \mathcal{P}_f[\boldsymbol{x}] \, \mathcal{D}[\boldsymbol{x}] \\
&= \int \prod_{k=0}^{K-1} \frac{f_{k+1}(x_k)}{f_k(x_k)} \times \prod_{k=1}^{K-1} T_k(x_k|x_{k-1}) \, p_0(x_0) \, dx_0 \cdots dx_{K-1} \\
&= \frac{1}{Z_0} \int f_K(x_{K-1}) \prod_{k=1}^{K-1} \frac{T_k(x_k|x_{k-1}) \, f_k(x_{k-1})}{f_k(x_k)} \, dx_0 \cdots dx_{K-1} \\
&= \frac{Z_K}{Z_0} \int p_K(x_{K-1}) \prod_{k=1}^{K-1} \frac{T_k(x_k|x_{k-1}) \, p_k(x_{k-1})}{p_k(x_k)} \, dx_0 \cdots dx_{K-1} \\
&= \frac{Z_K}{Z_0} \int p_K(x_{K-1}) \, dx_{K-1} \\
&= Z
\end{aligned}
\tag{2}
$$

where we used

$$\int \frac{T_k(x_k|x_{k-1})\, p_k(x_{k-1})}{p_k(x_k)}\, dx_{k-1} = \frac{1}{p_k(x_k)} \int T_k(x_k|x_{k-1})\, p_k(x_{k-1})\, dx_{k-1} = \frac{p_k(x_k)}{p_k(x_k)} = 1$$

We can also use the sampled paths $\boldsymbol{x}^{(i)}$ to approximate the target distribution, because

$$p_K(x) = \frac{1}{Z}\, \langle T_K(x|x_{K-1})\, w \rangle_f$$

which follows directly from the second last expression in equations (2). For each generated path $\boldsymbol{x}^{(i)}$ we have to generate just one more state by drawing from $x_K^{(i)} \sim T_K(x|x_{K-1}^{(i)})$. These samples can then be used to approximate the target by

$$p_K(x) \approx \frac{\sum_i w^{(i)}\, \delta\big(x - x_K^{(i)}\big)}{\sum_i w^{(i)}}\ .$$

## 1.2 Detailed fluctuation theorem

The detailed fluctuation theorem [3, 4] follows from comparing the probabilities of generating $\boldsymbol{x}$ in a forward and reverse simulation:

$$
\begin{aligned}
\frac{\mathcal{P}_f[\boldsymbol{x}]}{\mathcal{P}_r[\boldsymbol{x}]} &= \frac{T_{K-1}(x_{K-1}|x_{K-2}) \cdots T_1(x_1|x_0)\, p_0(x_0)}{T_1(x_0|x_1) \cdots T_{K-1}(x_{K-2}|x_{K-1})\, p_K(x_{K-1})} \\
&= \frac{p_0(x_0)}{p_K(x_{K-1})} \prod_{k=1}^{K-1} \frac{T_k(x_k|x_{k-1})}{T_k(x_{k-1}|x_k)} \\
&= \frac{p_0(x_0)}{p_K(x_{K-1})} \prod_{k=1}^{K-1} \frac{p_k(x_k)}{p_k(x_{k-1})} \\
&= \frac{Z_K}{Z_0} \prod_{k=0}^{K-1} \frac{f_k(x_k)}{f_{k+1}(x_k)} \\
&= \frac{Z}{w[\boldsymbol{x}]} \\
&= \exp\{\mathcal{W}[\boldsymbol{x}] - \Delta F\}
\end{aligned}
\tag{3}
$$

where detailed balance was assumed in the third equation. One interpretation of the detailed fluctuation theorem is that it gives the importance weights when a forward simulation is used to generate samples of the reverse ensemble and vice versa [2]. From a physical perspective the fluctuation theorem expresses microscopic reversibility [3, 4].

## 1.3 Relation to thermodynamic integration

According to inequalities (7) from the main text, $\langle \log w \rangle_f = -\langle W \rangle_f$ provides a lower bound of the log evidence. In case of thermal sampling of a Bayesian model, we have:

$$\langle W \rangle_f = \sum_{k=0}^{K-1} (\beta_{k+1} - \beta_k)\langle E \rangle_{q_k}$$

which approaches

$$\langle W \rangle_f \to \sum_{k=0}^{K-1} (\beta_{k+1} - \beta_k)\langle E \rangle_{p_k}$$

for large $K$ and/or large $N$ since, $q_k \to p_k$.

Recall that thermodynamic integration (TI) is based on the identity [5]:

$$\log Z = -\int_0^1 \langle E \rangle_\beta\, d\beta$$

where $\langle \cdot \rangle_\beta$ are *equilibrium* averages. For a finitely spaced inverse temperature schedule, we can approximate the TI integral by

$$\log Z \approx -\sum_k (\beta_{k+1} - \beta_k) \langle E \rangle_{p_k}$$

because $\langle \cdot \rangle_{\beta_k} = \langle \cdot \rangle_{p_k}$. The approximate formula is identical to the average work. That is, for large $K$ and/or large $N$ the lower bound $-\langle W \rangle_f$ approaches the log evidence and becomes identical to thermodynamic integration.

## 2 Gaussian kernel

This section provides background information about the toy model used in sections 4 and 5 of the main text. Let's define the normal distribution as

$$\mathcal{N}(x; \mu, \sigma^2) = \frac{1}{\sqrt{2\pi\sigma^2}} \exp\left\{-\frac{1}{2\sigma^2}(x - \mu)^2\right\}$$

with mean $\mu$ and standard deviation $\sigma > 0$. The following convolution theorem holds for Gaussian distributions:

$$\int \mathcal{N}(x; \mu_1, \sigma_1^2) \mathcal{N}(x; \mu_2, \sigma_2^2)\, dx = \mathcal{N}(\mu_1; \mu_2, \sigma_1^2 + \sigma_2^2).$$

Furthermore we have:

$$\mathcal{N}(x; a + bx', c) = \frac{1}{|b|} \mathcal{N}(x'; (x - a)/b, c/b^2)$$

Let us now consider the following transition kernel which involves three parameters $a, b, c$:

$$T(x|x') = \mathcal{N}(x; a + b\,x', c).$$

Then the action of the kernel on a normal distribution with parameters $\mu, \sigma$ is:

$$
\begin{aligned}
\int T(x|x') \mathcal{N}(x'; \mu, \sigma^2)\, dx &= \int \frac{1}{|b|} \mathcal{N}(x'; (x - a)/b, c/b^2) \mathcal{N}(x'; \mu, \sigma^2)\, dx' \\
&= \frac{1}{|b|} \mathcal{N}(\mu; (x - a)/b, c/b^2 + \sigma^2) \\
&= \mathcal{N}(x; a + b\mu, c + b^2\sigma^2).
\end{aligned}
$$

We choose $a, b, c$ such that $\mathcal{N}(x; \mu, \sigma^2)$ is the invariant distribution of $T(x|x')$:

$$a = (1 - b)\mu, \quad c = (1 - b^2)\sigma^2.$$

Then for any $\tau \in [0, 1]$

$$T(x|x') = \mathcal{N}(x; \tau y + (1 - \tau)\mu, (1 - \tau^2)\sigma^2) \tag{4}$$

has the desired stationary distribution. The parameter $\tau$ determines how quickly the Markov chain generated by $T$ converges to the stationary distribution. For $\tau = 0$, the convergence is immediate; for $\tau \to 1$, the convergence becomes infinitely slow.

The composition of two transition kernels with parameters $\tau_1, \mu_1, \sigma_1$ and $\tau_2, \mu_2, \sigma_2$ results in a new kernel with parameters:

$$
\begin{aligned}
\tau &= \tau_1 \tau_2 \\
\mu &= \frac{1}{1 - \tau_1 \tau_2}\left[(1 - \tau_1)\mu_1 + \tau_1(1 - \tau_2)\mu_2\right] \\
\sigma^2 &= \frac{1}{1 - (\tau_1 \tau_2)^2}\left[(1 - \tau_1^2)\sigma_1^2 + \tau_1^2(1 - \tau_2^2)\sigma_2^2\right].
\end{aligned}
$$

Repeated composition of the same transition kernel gives the $n$-th power

$$T^n(x|x') = \mathcal{N}(x; (1 - \tau^n)\mu + \tau^n x', (1 - \tau^{2n})\sigma^2)$$

that is, we simply have to raise $\tau$ to its $n$-th power: $T_\tau^n = T_{\tau^n}$. If we let $n \to \infty$, $\tau^n \to 0$ and $T^n(x|x') \to \mathcal{N}(x; \mu, \sigma^2)$ as it should.

## 3 Bennett's acceptance ratio

For two work distributions $p_i(W) = q_i(W)/c_i$ $(i = 0, 1)$, we have

$$\int h(W) \, q_0(W) \, q_1(W) \, dW = c_0 \langle hq_1 \rangle_0 = c_1 \langle hq_0 \rangle_1$$

where $h$ is a general function of the work. Therefore the ratio

$$r \equiv \frac{\langle hq_0 \rangle_1}{\langle hq_1 \rangle_0}$$

is an estimator of the ratio of the normalizing constants $c_1/c_0$. Assuming we have equally many samples $W_{ij}$ from $p_i(W)$, the sample version of the ratio estimator is

$$\hat{r} = \frac{\sum_j h(W_{1j}) \, q_0(W_{1j})}{\sum_j h(W_{0j}) \, q_1(W_{0j})}$$

In [6, 7, 5], it is shown that the relative mean squared error

$$\left\langle \frac{(r - \hat{r})^2}{r^2} \right\rangle$$

is minimized for

$$h(W) \propto \frac{1}{p_0(W) + p_1(W)}$$

resulting in an implicit estimator, because $p_i$ depends on $c_i$. Therefore, we have:

$$\hat{r} = \frac{\sum_j \frac{q_0(W_{1j})}{p_0(W_{1j}) + p_1(W_{1j})}}{\sum_j \frac{q_1(W_{0j})}{p_0(W_{0j}) + p_1(W_{0j})}} = \frac{\sum_j \frac{q_0(W_{1j})}{q_0(W_{1j}) + r \, q_1(W_{1j})}}{\sum_j \frac{q_1(W_{0j})}{q_0(W_{0j}) + r \, q_1(W_{0j})}} \, .$$

According to Crooks' fluctuation theorem, we have $q_0 \propto p_f$ and $q_1 \propto p_f e^{-W}$, resulting in the implicit equation

$$\hat{r} = \frac{\sum_j \frac{1}{1 + \hat{r} \, \exp\{-W_{1j}\}}}{\sum_j \frac{1}{\hat{r} + \exp\{W_{0j}\}}} = \hat{r} \times \frac{\sum_j \frac{1}{1 + \hat{r} \, \exp\{-W_{1j}\}}}{\sum_j \frac{1}{1 + \hat{r}^{-1} \exp\{W_{0j}\}}} \, .$$

Identifying simulation 0/1 with the forward/reverse simulation gives:

$$\hat{r} \leftarrow \hat{r} \times \frac{\sum_i \frac{1}{1 + \hat{r} \, \exp\{-W_r^{(i)}\}}}{\sum_i \frac{1}{1 + \hat{r}^{-1} \exp\{W_f^{(i)}\}}} \, .$$

Now $r = c_1/c_0 = 1/Z$, resulting in the multiplicative update:

$$\hat{Z} \leftarrow \hat{Z} \times \frac{\sum_i \frac{1}{1 + \hat{Z} \, \exp\{W_f^{(i)}\}}}{\sum_i \frac{1}{1 + \hat{Z}^{-1} \exp\{-W_r^{(i)}\}}} \tag{5}$$

which are the iterations used to compute the BAR estimator.

## 4 Histogram estimator

In DOS estimation [8, 9], we want to reconstruct the density of states $g(E)$ from energy samples that we generated according to $E_i \sim g(E) \, e^{-\beta_i E}/Z(\beta_i)$. We use the following analogy to use DOS estimation algorithms for the estimation of the work distribution $p_f$:

$$E \leftrightarrow W, \quad g(E) \leftrightarrow p_f(W), \quad \beta \in \{0, 1\} \, .$$

According to Crooks' fluctuation theorem, we have:

$$W_f^{(i)} \sim p_f(W), \quad W_r^{(i)} \sim p_f(W) e^{-W}/Z \, .$$

We want to estimate $p_f$ from all samples $W = \{W_f^{(1)}, \ldots\} \cup \{W_r^{(1)}, \ldots\}$ where $W_j$ are the elements of the joint set. We define the normalization constant of the work distribution:

$$c(\alpha) = \int e^{-\alpha W} p_f(W)\, dW$$

then the evidence is $Z = c(1)/c(0)$ according to Crooks' fluctuation theorem.

Given $p_f$, the likelihood of generating $W_f^{(i)}, W_r^{(i)}$ is:

$$\mathcal{L}[p_f] = \sum_j \log p_f(W_j) - N_f \log c(0) - N_r \log c(1) .$$

The maximum likelihood estimator is obtained by setting the functional derivative

$$\frac{\delta \mathcal{L}[p_f]}{\delta\, p_f(W)} = \frac{\sum_j \delta(W - W_j)}{p_f(W)} - \frac{N_f}{c(0)} - \frac{N_r}{c(1)} e^{-W}$$

to zero, which gives the implicit equation

$$\hat{p}_f(W) = \frac{h(W)}{\frac{N_f}{c(0)} + \frac{N_r}{c(1)} e^{-W}}$$

where $h(W) = \sum_j \delta(W - W_j)$ is the histogram of all simulated work values. This is an implicit equation, since $c(0)$ and $c(1)$ depend on $p_f$. We have:

$$\hat{c}(\alpha) = \int \hat{p}_f(W)\, dW = \sum_j \frac{e^{-\alpha W_j}}{\frac{N_f}{\hat{c}(0)} + \frac{N_r}{\hat{c}(1)} e^{-W_j}} .$$

We can show that by iterating these equations, we obtain the unique maximum likelihood estimate of $p_f$ and $c(\alpha)$ [8]. After convergence of the iterations, the maximum likelihood estimate of $p_f$ is:

$$\hat{p}_f(W) = \sum_j p_j\, \delta(W - W_j) \quad \text{with} \quad p_j \propto \frac{1}{\frac{N_f}{\hat{c}(0)} + \frac{N_r}{\hat{c}(1)} e^{-W_j}} .$$

If we are only interested in the evidence, we have to cycle over the following iterations:

$$\hat{Z} = \frac{\sum_j \frac{e^{-W_j}}{\frac{N_f}{N_r} \hat{Z} + e^{-W_j}}}{\sum_j \frac{1}{\frac{N_f}{N_r} \hat{Z} + e^{-W_j}}}$$

For $N_f = N_r$, this equation simplifies to

$$\hat{Z} = \frac{\sum_j \frac{1}{1 + \hat{Z} e^{W_j}}}{\sum_j \frac{1}{\hat{Z} + e^{-W_j}}}$$

A similar equation can be obtained from BAR:

$$\hat{Z} = \frac{\sum_i \frac{1}{1 + \hat{Z}\, \exp\{W_f^{(i)}\}}}{\sum_i \frac{1}{\hat{Z} + \exp\{-W_r^{(i)}\}}} .$$

Following [9], we can also derive a Gibbs sampler for $p_f$ and $c(\alpha)$:

$$p_j \sim \mathcal{G}\big(1, a_0 + a_1\, \exp\{-W_j\}\big)$$

$$a_0 \sim \mathcal{G}\Big(N_f, \sum_j p_j\Big) \tag{6}$$

$$a_1 \sim \mathcal{G}\Big(N_r, \sum_j p_j \exp\{-W_j\}\Big)$$

where the interpretation of the auxiliary parameters is $a_i = N_i/c_i$.

# 5 Sequential Monte Carlo

Instead of using an inverse temperature we define:

$$f_k(x) = \pi(x) \prod_{l=1}^{k} p(y_l|x, M), \ f_0(x) = \pi(x)$$

assuming that the data are independent. The weight of an entire path is:

$$w[\boldsymbol{x}] = \prod_{k=0}^{K-1} \frac{f_{k+1}(x_k)}{f_k(x_k)} = \prod_{k=0}^{K-1} p(y_{k+1}|x_k, M)$$

where $K$ is the number of data.