[Reviews · NeurIPS 2017]

Reviewer 1



# Summary of the paper The paper studies new estimators of the evidence of a model based on ideas from statistical physics. # Summary of the review The paper introduces simple methods to estimate the evidence of a model, that originate in the statistical physics literature. The methods seem interesting and novel (at least in the stats/ML literature). My main concerns are that the paper 1) lacks in clarity for an ML audience, and 2) lacks a fair comparison to the state-of-the art. In particular, is the contribution a mathematical curiosity, or is there an applicative setting in which it is the method of choice? # Major comments - Section 2: Sequential Monte Carlo samplers [Del Moral, Doucet, Jasra, JRSSB 2006] should also appear here, as they give extremely competitive evidence estimators. In particular, they allow more design choices than AIS. - Section 3: this section could gain in clarity and structure. Maybe you could isolate the important results as propositions. It also reads strangely to an ML reader, as the authors give lots of names and abbreviations (CFT, JE) for results that seem to be derivable in a few lines. Are these names really standard? - L91: this paragraph about merits and pitfalls of JE is unclear to me. - Section 5: as for Section 3, this section could gain in structure. For example, I would have liked the pseudocode of the main algorithm to be isolated from the main text, and I would have liked the BAR and the histogram methods to be clearly separated. After reading this section, it wasn't clear to me what the whole estimation procedure was, and what buttons you were going to tweak in the experimental Section 6. - Section 6.1: why do you start the reverse paths from fixed extremal states? I thought you needed them to be samples from the target distribution, or did I misunderstand something? In this small Ising case, perfect sampling would even be available via monotone coupling from the past. - Section 6.4: this is a typical example where SMC samplers would shine. - Section 6: overall, you do not mention in which setting your estimators would be state-of-the-art. For instance, the ground truth is actually found via parallel tempering. Why and when should I use your estimators and not parallel tempering? # Minor comments - Eqn 4: what is D[x]? I guess it's the d(path) in the integral, but since the notation is nonstandard, it should be explained. - L97 "a key feature" - L103 "then"

Reviewer 2



The authors provide a very thorough review of the use of non equilibrium fluctuation theorems from the statistical physics literature for the purpose of estimating the marginal likelihood of statistical models. In particular, two approaches are introduced from the physics literature that to my knowledge haven’t been used in the field of statistics/machine learning. I thought this was an good paper. It is clearly written and ties together many ideas that form the basis of a number of different but related marginal likelihood estimators. The authors explain the BAR and histogram estimator which I haven’t seen used in statistics and machine learning before. There is also a nice selection of simulation studies on some non-trivial statistical models. My only comment is that the authors should be careful about how they phrase their contribution, particularly their use of the word “new” - they should make clear that these are estimators that have already been published in the physics literature, but aren’t commonly used in statistics and machine learning. Aside from this minor issue, I think this paper makes a useful contribution by sharing knowledge between different fields.

Reviewer 3



This paper discusses tackling marginal likelihood estimation, a central problem in Bayesian model selection, using insights provided by so-called nonequilibrium fluctuation theorems. While Jarzynksi's Equality (JE) allows a natural sequential Monte Carlo estimator of log Z, the estimator can be high-variance if the resultant sequence of nonequilibrium distributions have insufficient overlap, which is a common problem for complex, multimodal problems. The fluctuation theorem allows (under generally non-realistic assumptions; more shortly) a bounding of the estimate no matter how fast the annealing schedule is. In addition to providing bounds, the reverse work distribution can be incorporated into estimators for log Z itself that are (hopefully) an improvement over the one provided by JE. Experiments are run on various statistical models. I found this to be an interesting paper and enjoyed reading it. My main reservations are concerned with the feasibility of simulating the reverse process in most situations that are practically-challenging. It appears that the authors do recognize this limitation, but provide a brief and not overly satisfying discussion at the end of p.4. Most of the experimental results focus on problems with a simple low-energy structure (the ferromagnetic Ising and Potts models, and mixture models with 3 components) The RBM example is a bit more intriguing, but when trained on MNIST, one can reasonably infer the existence of a relatively small number of modes. In general, the method assumes access to an oracle providing a perfectly-trained model, where the data points do indeed represent the modes of the distribution (1000 sweeps of Gibbs sampling at low temperature will move only locally and cannot be expected to move the state too far from the training points.) While the agreement of the current method and PT log Z results in this case is encouraging, the fit of the model is generally difficult to verify in practice- indeed that's the reason for conducting the simulation in the first place. To phrase the objections in another way, optimization is generally a very different problem from sampling, as represented by the distinction between problems in NP and those in #P. There exist several interesting types of problems for which computing the ground state energy is tractable, but counting the number of such minima (obtaining the ground state entropy) is not. Generating samples uniformly from the ground state distribution, as would be required for the reverse methodologies, will then be problematic. Another issue was that on first reading, it was not immediately clear what the paper's contributions are; presumably, it is the combined forward/reverse weighted estimator presented on page 5? The remaining ones have been previously introduced in the literature. While valid, I'm ambivalent as to whether this contribution warrants an entire NIPS paper. Certainly, the connection between marginal likelihood and free energy estimation is widely-known among Bayesian computational statisticians. Despite these misgivings, I still found the paper to be quite cool and am encouraged to see progress in this research area. Technical correctness: ------------------- The paper appears to be technically correct to my knowledge. Clarity: ------- The paper is quite well-written and easy to follow for someone familiar with Sequential Monte Carlo and Bayesian statistics. The presentation could be better-adapted to the NIPS context, however, as physics-flavored notions such as "work" may seem distant to a machine learning audience. Specific issues: --------------- 252-253: In what sense is simulation of marginal RBM 3x faster than that of the joint? The joint simulation can be parallelized and simulated on a GPU due to the conditional independence structure of the RBM. The statement should be clarified.